# Sensory Processing Sensitivity and Its Relation to Exercise Behavior and Preferred Exercise Intensity

**DOI:** 10.3390/jfmk10010018

**Published:** 2025-01-03

**Authors:** Benjamin Lee Webb

**Affiliations:** Department of Applied Health, Southern Illinois University Edwardsville, Edwardsville, IL 62026, USA; benjweb@siue.edu

**Keywords:** sensory processing sensitivity, personality, physical activity, physical health, exercise promotion

## Abstract

**Background/Objectives:** Regular exercise is important for overall health, yet exercise participation in the United States remains low. Exercise promotion depends on identifying factors such as personality that might influence exercise participation. Sensory processing sensitivity (SPS), a personality trait described as the tendency to deeply process environmental stimuli, is a psychological factor that may influence exercise participation. The purpose of this cross-sectional study was to examine relationships among SPS, exercise behavior, and preferred exercise intensity. **Methods:** Participants (*N* = 320) were college students and employees who completed the 12-Item Highly Sensitive Person Scale, the Godin Leisure-Time Exercise Questionnaire, and a question related to preferred exercise intensity. **Results:** Participants’ ages ranged from 18 to 70 years (*M* = 39.36, *SD* = 15.15), and they were mostly female (69.6%). Most participants were physically active (77.5%). Mean SPS scores were not significantly different between active (*M* = 50.2, *SD* = 10.9) and insufficiently active (*M* = 51.4, *SD* = 9.97) participants; however, post hoc analysis revealed that the mean increase in SPS score from preference for vigorous intensity to light intensity (5.18, 95% CI [0.13, 10.2]) was statistically significant (*p* = 0.043). **Conclusions:** Exercise preferences are an important consideration for exercise adoption and adherence; thus, these findings have practical implications for exercise promotion, especially for individuals who score higher in SPS.

## 1. Introduction

Regular participation in exercise, a form of leisure-time physical activity, is associated with numerous physical and mental health benefits [1]; however, nearly one-quarter of U.S. adults state that they do not participate in any leisure-time physical activity [2]. There is also evidence to suggest that exercise participation has not improved since the 2008 physical activity guidelines were published [3]. Being insufficiently active increases the risk of heart disease, stroke, hypertension, type 2 diabetes, and many other chronic health conditions [1]. Therefore, developing interventions to promote exercise remains a public health priority [4] whose success depends on identifying factors associated with exercise participation.

A growing body of evidence supports the influence of personality on exercise participation. For example, a systematic review and meta-analysis conducted by Wilson and Dishman provides support for the influence of personality traits on exercise participation, with their findings confirming associations across multiple studies and populations [5]. Their research demonstrated that individuals low in neuroticism and high in conscientiousness and extraversion, all factors from the Five-Factor Model of personality [6], consistently reported greater amounts of physical activity. These associations indicate that there are patterns that are moderately heterogeneous across various sample and research design characteristics. While earlier studies provided foundational insights into these relationships [7,8], Wilson and Dishman systematically quantified and expanded upon this evidence [5]. To further inform the development of exercise interventions, it is important to explore additional personality traits that may be associated with exercise participation.

Building on the established links between personality traits and exercise behavior, it is worth examining Sensory Processing Sensitivity (SPS), another personality trait that may influence exercise participation. The SPS trait is present to varying degrees in all humans [9,10]. The SPS trait is described as the degree to which individuals deeply process environmental stimuli [11]. According to Aron and Aron [11], individuals with high SPS often report “sensitivity to subtleties, the arts, caffeine, hunger, pain, change, overstimulation, strong sensory input, others’ moods, violence in the media, and being observed by others”. Many of these characteristics, such as sensitivity to subtleties, pain, overstimulation, strong sensory input, and being observed by others, have practical implications for exercise promotion. For example, individuals with higher pain sensitivity have reported greater perceived pain following exercise [12], which could serve as a barrier to exercise adherence [13,14]. The limited research on SPS and exercise suggests that individuals with higher SPS participate in less exercise. For instance, Yano and Oishi examined associations between SPS, its subcomponents, depressive tendencies, and frequency of physical exercise among Japanese university students [15]. Of relevance to the current study, they found that low sensory threshold and ease of excitation, both subcomponents of SPS, were negatively correlated with frequency of physical exercise. A third subcomponent of SPS, aesthetic sensitivity, showed no significant correlation with exercise frequency [15]. Based on these findings, we expected that SPS scores would be significantly higher for insufficiently active individuals compared to active individuals.

Individuals with high SPS often experience greater emotional reactivity when exposed to internal or external stimuli [11,16,17,18]. An internal stimulus related to exercise is intensity, often described as light, moderate, or vigorous, depending on the amount of energy required for the activity [1]. As intensity increases, there are typically increases in heart rate [19], breathing rate [20], and perspiration [21]. There is also an accumulation of metabolic byproducts that may contribute to muscle fatigue [22] and exercise-induced pain [23,24]. Depending on the individual, these physical sensations can be evaluated positively or negatively. Previous research indicates that exceeding an individual’s preferred (i.e., self-selected) exercise intensity by even a small amount can lead to a decrease in pleasure and decreased adherence [25]. Individuals with high SPS may be sensitive to the strong sensory input associated with increasing exercise intensity; thus, we expected to find SPS scores would be significantly higher for individuals who prefer light-intensity exercise compared to those who prefer moderate- or vigorous-intensity exercise.

Although previous research indicates that individuals with higher SPS participate in less exercise [15], little is known regarding SPS and preferences for exercise intensity or environment. Therefore, the purpose of this cross-sectional study was twofold: (1) to confirm previous findings regarding the negative association between SPS and exercise behavior and (2) to examine associations between SPS and preferred exercise intensity.

## 2. Materials and Methods

### 2.1. Research Design

This observational study used a cross-sectional design to collect data from a convenience sample of participants using web-based questionnaires.

### 2.2. Participants and Procedures

Participants (*N* = 320) were employees and students enrolled at Southern Illinois University Edwardsville, a midwestern university in the U.S. Participants were recruited via email through the university’s listserv. Participants were required to provide consent to participate in the study before completing web-based questionnaires regarding their personality, exercise behavior, and exercise preferences. The ages of the participants ranged from 18–70 years (39.4 ± 15.2). The link to the web-based questionnaire was accessed by 424 individuals, with 336 completing the questionnaires (79% completion rate). Sixteen of the questionnaires were excluded from the analyses due to missing data on at least one of the key outcomes (SPS and exercise behavior). No data imputation was required on the remaining questionnaires. The Institutional Review Board at Southern Illinois University Edwardsville approved this study.

### 2.3. Measures

#### 2.3.1. Sensory Processing Sensitivity

The 12-item version of the Highly Sensitive Person (HSP) Scale was used to collect data on SPS [26]. The scale includes such items as, “Do you seem to be aware of subtleties in your environment?” and “Are you bothered by intense stimuli, like loud noises or chaotic scenes?” Each item uses a seven-point Likert-type scale ranging from 1 (Not at all) to 7 (Extremely), with a higher summary score indicative of higher SPS. Previous research indicates that the HSP scale offers a bi-factor solution for measuring SPS: three sub-scales and a general factor [9]. In accordance with the recommendations of Aron and Aron [27], the general factor (i.e., summary score for the 12 items) was used for this study. The scale had a high level of internal consistency, as determined by a Cronbach’s alpha of 0.781.

#### 2.3.2. Exercise Behavior

The Godin Leisure-Time Exercise Questionnaire (GLTEQ) was used to collect data on time spent in light, moderate, and strenuous exercise behaviors [28]. The GLTEQ measures frequency of light, moderate, and strenuous exercise lasting at least 15 min during the last 7 days. Participants are provided descriptions and examples of the various types of exercise for each intensity level. The frequency scores are multiplied by the corresponding Metabolic Equivalent of Task (MET) value (3, 5, and 9 for light, moderate, and vigorous intensity, respectively) and summed to obtain a leisure activity score. The leisure activity score uses arbitrary units to categorize people as insufficiently active/sedentary (<14 units), moderately active (14–23 units), or active (≥24 units) [28]. More recently, cutoffs for active (≥24) and insufficiently active (<24) have been established [29], which were used in this study. The GLTEQ been validated against objective measures of fitness (e.g., VO_2max_) [29,30] and has demonstrated validity in classifying healthy adults as active and insufficiently active [29].

#### 2.3.3. Preference for Exercise Intensity

Participants were asked to identify the intensity of exercise they preferred most from the following options: vigorous-intensity (i.e., activities where the heart beats rapidly, such as running, jogging, hockey, football, soccer, squash, basketball, cross country skiing, judo, roller skating, vigorous swimming, and vigorous long distance bicycling), moderate-intensity (i.e., activities that are not exhausting, such as fast walking, baseball, tennis, easy bicycling, volleyball, badminton, easy swimming, alpine skiing, popular and folk dancing), and light-intensity (i.e., activities that require minimal effort, such as yoga, archery, fishing from a river bank, bowling, horseshoes, golf, snow-mobiling, and easy walking). For consistency in terminology, the descriptions of three intensity levels were adapted from the GLTEQ [28].

#### 2.3.4. Demographics

To describe the sample, participants were asked to report their age (years) and gender (male, female, or an option to enter their gender identity).

### 2.4. Data Analysis

Descriptive statistics (e.g., mean, standard deviation, frequency, and percentage) are reported for all variables of interest. An independent-samples *t* test was used to examine whether mean SPS scores were significantly different between active and insufficiently active participants. One-way ANOVA was used to examine whether mean SPS scores differed significantly based on preferences for exercise intensity (vigorous, moderate, and light). Pairwise comparisons using Tukey’s HSD were conducted to determine which means differed significantly. Statistical analyses were conducted using IBM SPSS, Version 25.0 (Armonk, NY, USA). The significance level for all analyses was set at *p* < 0.05. A sample-size calculation using G*Power determined that, for one-way ANOVA with three groups (i.e., intensity preference), a sample size of 252 was needed to detect a medium effect size of 0.25 with a power of 0.95 at the 0.05 significance level [31].

## 3. Results

### 3.1. Description of the Sample

A description of all study variables is presented in Table 1. Most of the participants were active (77.5%) and female (69.6%), with a mean age of 39.7 years (± 15.1). A greater number of both active and insufficiently active participants preferred moderate-intensity exercise more than vigorous- or light-intensity exercise. An independent-samples *t*-test was performed to determine if there were differences in SPS score, leisure activity score, and age between active and insufficiently active participants. Importantly, the mean SPS score for active (50.2 ± 10.9) versus insufficiently active participants (51.4 ± 9.97) was not significantly different, *t*(318) = 0.84, *p* = 0.401, *d* = 0.11.

### 3.2. SPS and Preferred Exercise Intensity

One-way ANOVA was conducted to determine if the mean SPS score differed by preference for exercise intensity. Participants were classified into three groups based on intensity preference: vigorous (*n* = 93), moderate (*n* = 193), and light (*n* = 34). Data are presented as mean ± standard deviation. There was homogeneity of variances, as assessed by Levene’s test for equality of variances (*p* = 0.75). The mean SPS score was significantly different between at least two intensity preferences, *F*(2, 317) = 2.921, *p* < 0.05. The effect size, eta squared (η^2^), was 0.02, indicating a small effect of SPS on intensity preference. The SPS score increased from vigorous (49.1 ± 18.9) to moderate (50.5 ± 10.7) to light (54.2 ± 9.96), in that order. Tukey post hoc analysis revealed that the mean increase from vigorous to light (5.18, 95% CI [0.13, 10.2]) was statistically significant (*p* = 0.043). Mean SPS scores did not differ significantly between vigorous and moderate intensity (*p* = 0.522) or between moderate and light intensity (*p* = 0.151).

## 4. Discussion

The current study explored the relationships among SPS, exercise behavior, and preferences for exercise intensity. Contrary to our hypothesis, we found no significant difference in SPS scores between active and insufficiently active participants. However, there is evidence that preference for exercise intensity was associated with differences in SPS scores. Specifically, participants who preferred light-intensity exercise scored significantly higher in SPS compared to those who preferred vigorous-intensity exercise. These findings contradict prior research that has shown a negative correlation between SPS and frequency of exercise [15], suggesting that heightened sensitivity to sensory stimuli may not deter individuals from engaging in regular exercise. Unlike Yano and Oishi [15], who measured frequency of exercise behavior with no parameters for time or intensity, the current study measured frequency of exercise with parameters for both time and intensity. This distinction underscores the importance of considering both time and intensity when examining the relationship between SPS and exercise behavior, as these variables may reveal subtleties overlooked in previous research.

Our findings suggest that individuals with high SPS are more likely to prefer light-intensity exercise, which is overlooked in most PA guidelines [1,32]. However, research indicates that replacing sedentary behavior with light-intensity exercise can positively impact various health markers [33,34,35]. Exceeding an individual’s preferred exercise intensity has been shown to diminish positive affect [25], which may, in turn, reduce exercise adherence [36]. Therefore, initiating exercise programs at a light intensity for individuals with high SPS could enhance adherence while still delivering health benefits. This approach may be particularly beneficial for individuals with high SPS and low fitness levels, as they are more likely to experience negative affective responses (e.g., displeasure) to vigorous-intensity exercise, which can further hinder adherence [36,37]. Following a gradual progression to moderate or vigorous intensity as fitness improves, their preference for exercise intensity may change, provided the experiences they have at varying intensities are perceived primarily as positive, as suggested by experimental research [38,39]. A move toward more intense levels of exercise would allow them to experience additional health benefits. Further research is necessary to examine whether SPS is associated with lower fitness levels, as well as specific physiological or psychological responses to exercise that could influence preferred intensity levels. Additionally, exploring gradual exposure to exercise of varying intensities paired with coping strategies could be a promising direction for experimental research related to exercise and SPS.

### Limitations

While our study provides valuable insights into the relationship between SPS and exercise, several limitations warrant consideration. First, this was a cross-sectional study with a convenience sample of participants affiliated with a midwestern university. Additionally, most of the participants in this study reported that they were physically active, which is not representative of the general population. Thus, our findings do not establish causality and may have limited generalizability beyond this sample or region. Second, our reliance on self-reported exercise behaviors introduces potential biases, such as recall and social desirability, which could lead participants to misreport their activity levels. Whenever possible, future research should complement subjective measures of exercise with objective measures of exercise (e.g., accelerometers) to improve the accuracy of the data. Third, we did not account for participants’ prior experiences with exercise, including their exposure to different exercise intensities, which could impact current exercise behaviors and preferences. For instance, both observational [40] and experimental [9,41] research have found that individuals who score high in SPS have stronger reactions to positive and negative experiences than their low SPS counterparts. Thus, collecting data on previous exercise experiences would help clarify their influence on current behavior and preferences. Fourth, we used a single-item question to assess preferred exercise intensity. A validated measure, such as the preference scale from the Preference for and Tolerance of the Intensity of Exercise Questionnaire [42], would have provided more comprehensive data for our analyses. Lastly, we did not assess other personality traits, such as openness and neuroticism, which are associated with SPS and may also impact exercise behaviors. Incorporating additional personality factors could provide a more comprehensive understanding of how individual traits influence exercise behavior and preferences.

## 5. Conclusions

Our findings highlight the importance of tailoring exercise prescriptions to individual preferences, particularly for individuals with high SPS. Although higher SPS was not associated with lower activity levels, the preference for light-intensity exercise suggests a need to prioritize this intensity in initial exercise programs for this population. Incorporating light-intensity activities into physical activity guidelines and interventions could improve adherence and provide meaningful health benefits, especially for those with low fitness levels or negative affective responses to more intense exercise. Gradual progression to moderate or vigorous intensities as fitness improves could further improve health outcomes. Future research should examine whether targeted interventions based on preferred intensity can improve long-term adherence and overall well-being for highly-sensitive individuals.

## Figures and Tables

**Table 1 jfmk-10-00018-t001:** Descriptive statistics for all study variables (*N* = 320).

Variables	Active (*n* = 248)	Insufficiently Active (*n* = 72)	*t*-Test	Cohen’s *d*
	Freq (%)	Freq (%)		
SPS Summary Score	50.2 ± 10.9	51.4 ± 9.97	0.84	0.11
Leisure Activity Score	58.7 ± 29.0	11.2 ± 7.70	−13.7 **	−1.84
Preferred Exercise Intensity				
Light	10 (4.0)	24 (33.3)		
Moderate	153 (61.7)	40 (55.6)		
Vigorous	84 (34.3)	8 (11.1)		
Gender				
Female	171 (69.0)	51 (70.8)		
Male	77 (31.0)	20 (27.8)		
Age (years)	38.7 ± 14.7	41.8 ± 16.4	1.57	0.21

** *p* < 0.001 Note: SPS = sensory processing sensitivity.

## Data Availability

The original data presented in the study are openly available from FigShare at https://doi.org/10.6084/m9.figshare.27880455.

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
