# Peer review of "Sensory Processing Sensitivity and Its Relation to Exercise Behavior and Preferred Exercise Intensity"

_jfmk, 2025, doi:10.3390/jfmk10010018_

Round 1
Reviewer 1 Report
Comments and Suggestions for Authors
Your topic is important. My hope is my comments help you further this manuscript and your future research on this topic.
Abstract
Please provide the important statistics such as mean age, age range, % female, and your odds ratios.
Keywords
The idea of keywords is to use words not in your title to increase probability in searches. Please update to reflect the purpose of keywords. You have lots of good choices left such as leisure-time physical activity, physical health, and so on.
Introduction
Given this is 2024, your first paragraph of references seems very out-of-date. Please update at least the Tucker 2011.
I think Wilson and Dishman should lead off your second paragraph. The individual studies are handpicked when the meta-analysis is of course the best summary. The effects are small to very small. Thus, your information should reflect what the effects really are. They are important as any help in increasing/understanding PA behaviors is important. But handpicking studies do not reflect Wilson and Dishman. I heard Rod speak in 2016. He is right, even with small effects, if one conscientious exerciser calls a non-exerciser then you have one more exerciser in the world.
I think you need to tell us more about Yano et al. 2018.
Methods
You could tell the reader more about the Godin cutoffs for activity levels.
Line 123 – just descriptive as opposed to basic seems just fine. Descriptive information is what they are.
Why did you not use the PRETIE? I think you need to explain this as a limitation.
https://faculty.sites.iastate.edu/ekkekaki/pretie-q
Results
We need a table of the descriptive information by active and inactive classification.
Discussion
A section of limitations is needed.
Author Response
First, I would like to thank the reviewer for the helpful comments and suggestions. Having addressed them, I think the manuscript is in much better shape. Thank you for taking time to review my manuscript and for providing thoughtful comments.
Comment 1: Abstract: Please provide the important statistics such as mean age, age range, % female, and your odds ratios.
Response 1: I have made changes to the abstract based on your feedback, and that of other reviewers. This includes your recommendation to include some of the important statistics. (Page 1, lines 6-21).
Comment 2: Keywords: The idea of keywords is to use words not in your title to increase probability in searches. Please update to reflect the purpose of keywords. You have lots of good choices left such as leisure-time physical activity, physical health, and so on.
Response 2: Thank you for the suggestion. I have amended the selection of keywords. (Page 1, lines 23-24).
Comment 3: Given this is 2024, your first paragraph of references seems very out-of-date. Please update at least the Tucker 2011.
Response 3: I agree with your assessment. Unfortunately, Tucker et al. is the last group I can find that examined a subset of the population using objective methods; therefore, I have removed this citation and updated the information to include only the most recent data from the CDC for self-reported physical activity. (Page 1, lines 30-32).
Comment 4: I think Wilson and Dishman should lead off your second paragraph. The individual studies are handpicked when the meta-analysis is of course the best summary. The effects are small to very small. Thus, your information should reflect what the effects really are. They are important as any help in increasing/understanding PA behaviors is important. But handpicking studies do not reflect Wilson and Dishman. I heard Rod speak in 2016. He is right, even with small effects, if one conscientious exerciser calls a non-exerciser then you have one more exerciser in the world.
Response 4: Thank you for the suggestions. I made many revisions I think this improved this section of the introduction. (Page 1-2, lines 39-48).
Comment 5: I think you need to tell us more about Yano et al. 2018.
Response 5: Agreed. I have elaborated on Yano and Oishi's paper, which helps the reader to learn more about their study's purpose and where my study differed in some ways, especially when I address the differences in our measures of exercise behavior in the discussion. (Page 2, lines 69-74).
Comment 6: Methods: You could tell the reader more about the Godin cutoffs for activity levels.
Response 6: Thank you for the suggestion. I was too concise in my previous description, and should not assume all readers will be familiar with the instrument. I have elaborated on the GLTEQ. (Page 3, lines 130-141).
Comment 7: Line 123 – just descriptive as opposed to basic seems just fine. Descriptive information is what they are.
Response 7: Thank you. This has been revised.
Comment 8: Why did you not use the PRETIE? I think you need to explain this as a limitation.
Response 8: Thank you for informing me of this instrument. I am quite familiar with Dr. Ekkekakis' work on affective responses to exercise, so I am embarrassed I was unaware of this instrument. It certainly would have strengthened the methods of my study, and I have acknowledged this in the limitations. (Page 6, lines 260-264).
Comment 9: We need a table of the descriptive information by active and inactive classification.
Response 9: I have revised much of the results section based on changes to the data analyses, including adding a descriptive table with participants classified as active and insufficiently active. (Page 4, Table 1, lines 180-181).
Comment 10: A section of limitations is needed.
Response 10: Although the limitations were there in the original manuscript, the subheading was not. I have added the subheading (Page 6, line 245) and revised some of the limitations based on your feedback (see response 8).
Reviewer 2 Report
Comments and Suggestions for Authors
I have read with interest the manuscript "Associations between Sensory Processing Sensitivity, Exercise Behavior, and Preferred Exercise Intensity." Although the idea of the study is relevant, there are many issues that need to be resolved before the article can be considered for publication.
1. The citation method is not MDPI compliant, which makes it difficult to find references to the literature. The authors use APA style in the text, but in the list they number the subsequent items in alphabetical order. This requires a thorough change, and the authors should familiarize themselves with the journal's requirements before submitting their paper.
2. The authors conducted the study in the academic environment of one university. However, it is not clear whether the participants were students or academic staff. It is impossible to repeat this study because of the lack of important information on the characteristics of the sample. Please add information on the prevalence of students in each year of study, whether they are bachelor's or master's students, and what fields of study they represented. First of all, it is not known whether they are currently participants in sports or academic clubs and what disciplines they represent. How long have they been in sports? This is necessary to control the sports activity in the study group.
3. How was sample size determined? Please provide the results of power analysis.
4. The Highly Sensitive Person (HSP) Scale consists of 12 items and the responses are expressed on a 7-point Likert scale. This means that the score is expressed on a continuous scale, not a categorical one. Therefore, the reliability of this scale (e.g. Cronbach's alpha) should be reported and the statistical analyses for testing hypotheses should be changed. I suggest using the Student's t-test for the Activity Category factor and one-way ANOVA for the Intensity Preference factor. Of course, appropriate post-hoc tests and effect sizes should be added.
5. Discussion and limitation sections should be expanded to include new data and unresolved issues regarding the demographic description of the group.
6. the separate "Conclusion" section should be added. Please add more future directions and concrete practical implications from this study.
Author Response
Thank you for the helpful feedback. Although I was not able to attend to all of your suggestions, I was able to take care of most of what you suggested. I think the methods and results are improved based on your feedback, and I am very appreciative of the time you took to offer your guidance.
Comment 1: The citation method is not MDPI compliant, which makes it difficult to find references to the literature. The authors use APA style in the text, but in the list they number the subsequent items in alphabetical order. This requires a thorough change, and the authors should familiarize themselves with the journal's requirements before submitting their paper.
Response 1: I apologize for the confusing formatting. MDPA now permits authors to submit their manuscript in its original format (e.g., APA), and then it is placed into their template for the reviewers. This resulted in my manuscript appearing to be in two different citation styles. The citation style has been updated to reflect the style preferred by MDPI.
Comment 2: The authors conducted the study in the academic environment of one university. However, it is not clear whether the participants were students or academic staff. It is impossible to repeat this study because of the lack of important information on the characteristics of the sample. Please add information on the prevalence of students in each year of study, whether they are bachelor's or master's students, and what fields of study they represented. First of all, it is not known whether they are currently participants in sports or academic clubs and what disciplines they represent. How long have they been in sports? This is necessary to control the sports activity in the study group.
Response 2: Thank you for this observation. I am not able to gather the suggested information now that the study is complete. Although having this additional information would be valuable for a more comprehensive description of the sample, and for additional statistical modeling, I had no reason to assume SPS would vary based on the factors mentioned. There is some evidence that it might vary based on gender, but there is no consensus on that, either.
Comment 3: How was sample size determined? Please provide the results of power analysis.
Response 3: Thank you for this suggestion. I have added the results of the sample-size calculation based on your suggestions regarding rerunning the statistics using t tests and an ANOVA. (Page 4, lines 168-171).
Comment 4: The Highly Sensitive Person (HSP) Scale consists of 12 items and the responses are expressed on a 7-point Likert scale. This means that the score is expressed on a continuous scale, not a categorical one. Therefore, the reliability of this scale (e.g. Cronbach's alpha) should be reported and the statistical analyses for testing hypotheses should be changed. I suggest using the Student's t-test for the Activity Category factor and one-way ANOVA for the Intensity Preference factor. Of course, appropriate post-hoc tests and effect sizes should be added.
Response 4: These were excellent suggestions that reminded me to be more diligent in my data analysis plan. I have added information regarding the reliability of the HSP scale items (Page 3, lines 127-128), as well as revised the data analysis plan (Page 4, lines 158-170), results (Page 4-5, Lines 172-202), and discussion of the results (Page 6) based on your recommendations. I think this has substantially improved the methods of the study.
Comment 5: Discussion and limitation sections should be expanded to include new data and unresolved issues regarding the demographic description of the group.
Response 5: Thank you for the suggestion. These have been expanded on (Page 6).
Comment 6: the separate "Conclusion" section should be added. Please add more future directions and concrete practical implications from this study.
Response 6: Thank you. This has been added and improved upon based on the new data analyses.
Reviewer 3 Report
Comments and Suggestions for Authors
Dear author, despite the quality of the present work, I consider it essential that the present concerns could be addressed:
Line 22: Despite the exercise rate being low, it is essential to describe what it means. More specific numbers about it are needed.
Line 28-38: The author identified some trait of personality with exercise participation. However, it is also interesting that authors could identify if there is any personality trait positively linked to low exercise participation rate.
Line 39: The author should make a better transition between paragraphs. Moreover, in this case (lines 38–39), it is necessary to give us information that leads the author to choose the SPS in detriment of other traits. Beyond that information given to us by the author further, maybe a conceptual framework of the personality traits could become more subtle in this transition.
Line 68: Were not identified in the populations of the previous studies in the field. It is important since we don't know about the study population and how it can fill the gap in area.
An analysis of sample size is required.
The author also should indicate whether scales are adequate to US reality.
An analysis of data normality should be indicated.
Line 170-173: It could be true. However, it was identified that increasing SPS raises the likelihood of lower exercise intensity in comparison with higher exercise intensities... Therefore, if the subject's SPS perception remains higher, this sentence couldn't make sense.
As study's limitations, the author should include the fact that the study's sample was chosen by convenience and the fact that these subjects have a physical activity behavior distinct from the common population, since mainly of them are physically active.
The main conclusion of the study should appear in a proper section.
I would like to thank the authors for their work and hope that my feedback can improve the quality of this manuscript.
Author Response
First and foremost, I would like to thank you for taking the time to review my manuscript, and for providing thoughtful suggestions on how to improve its quality. I hope I have adequately addressed your concerns.
Comment 1: Line 22: Despite the exercise rate being low, it is essential to describe what it means. More specific numbers about it are needed.
Response 1: Thank you for the feedback. If I understand you correctly, you want elaboration on why being insufficiently active could be problematic. The opening sentence implies why not getting enough activity could be problematic, but I have added a bit more context to emphasize this point. I have revised this section based on comments from other reviewers, as well. (Page 1, lines 34-36).
Comment 2: Line 28-38: The author identified some trait of personality with exercise participation. However, it is also interesting that authors could identify if there is any personality trait positively linked to low exercise participation rate.
Response 2: This is a good question. If you consider that the research I reported on shows correlations between personality traits that run on a continuum (i.e., low and high), then we can conclude that an opposite score on the scale that measures the trait must correlate with less exercise. For example, someone who scores high in neuroticism is less likely to exercise, as would be someone who scores low in extraversion (i.e., introverts). There may be additional, less-studied traits associated with exercise behavior that I did not explore in the introduction, but I wanted to be selective for the sake of space.
Comment 3: Line 39: The author should make a better transition between paragraphs. Moreover, in this case (lines 38–39), it is necessary to give us information that leads the author to choose the SPS in detriment of other traits. Beyond that information given to us by the author further, maybe a conceptual framework of the personality traits could become more subtle in this transition.
Response 3: Thank you for the suggestion. Given the opportunity, I could try to smooth out the transitions even more. I have revised some of the introduction to help with the flow of the text, and have attempted to improve the transition to introducing the SPS trait.
Comment 4: Line 68: Were not identified in the populations of the previous studies in the field. It is important since we don't know about the study population and how it can fill the gap in area.
Response 4: Thank you for the suggestion. I have elaborated on Yano and Oichi's study for the reader. (Page 2, lines 70-78).
Comment 5: An analysis of sample size is required.
Response: Thank you. This has been added to the data analysis section. (Page 4, lines 168-171).
Comment 6: Line 170-173: It could be true. However, it was identified that increasing SPS raises the likelihood of lower exercise intensity in comparison with higher exercise intensities... Therefore, if the subject's SPS perception remains higher, this sentence couldn't make sense.
Response 6: Great observation. When writing the discussion, I had considered the same thing, especially considering SPS is thought to be a relatively stable trait with some variation depending on the situation and individual. I have amended this section of text to clarify that the experience of increasing intensity most likely needs to be perceived positively based on the limited experimental evidence there is. I've also added the need for experimental research regarding SPS and exercise to this paragraph, too.
Comment 7: As study's limitations, the author should include the fact that the study's sample was chosen by convenience and the fact that these subjects have a physical activity behavior distinct from the common population, since mainly of them are physically active.
Response 7: I agree and have revised the limitations to explicitly state these as limitations of the study.
Comment 8: The main conclusion of the study should appear in a proper section.
Response 8: Thank you for the suggestion. I have revised the manuscript to ensure all headings and subheadings are present.
Round 2
Reviewer 1 Report
Comments and Suggestions for Authors
Thank you for your revision. I wish you success in your future research.
Author Response
Comments 1: Thank you for your revision. I wish you success in your future research.
Response 1: Thank you once again for your prompt review and dedication to your discipline.
Reviewer 2 Report
Comments and Suggestions for Authors
I thank the authors for responding to my comments. Indeed, a significant part has been improved in this version of the manuscript. However, several problems remain unanswered, so I repeat my comments:
1. The lack of a description of the research group significantly reduces the quality of this research, and above all, makes it impossible to conduct replication of this study. It is not known what the structure of the research group is: what percentage were employees and students, what faculties they represented, what years of study. In addition to replication problems, this is also of significant importance for possible interventions or preventive activities in the academic setting. There are also no in-depth analyses using these data, so the value of the research is negligible. This problem should be discussed in detail in the "limitations" section.
2. The authors conducted a Student's t-test and a one-way ANOVA. The effect size (e.g. Cohen's d) should be added to Table 1 and discussed in the description of the results above the table. Also, the effect size for the ANOVA (e.g. partial eta-square) is missing. Please, fill these gaps, by reporting the effect size and its interpretation appropriately.
Author Response
Thank you once again for your prompt review of my revised manuscript. I appreciate your commitment to providing high-quality, critical evaluations of research reports. You have stimulated my thinking regarding areas of future research in SPS and exercise.
Comment 1: The lack of a description of the research group significantly reduces the quality of this research, and above all, makes it impossible to conduct replication of this study. It is not known what the structure of the research group is: what percentage were employees and students, what faculties they represented, what years of study. In addition to replication problems, this is also of significant importance for possible interventions or preventive activities in the academic setting. There are also no in-depth analyses using these data, so the value of the research is negligible. This problem should be discussed in detail in the "limitations" section.
Response 1: I appreciate the desire to collect as much demographic information as possible about a research sample to allow for a more robust description of the sample and analysis of the data, but I disagree with the assertion that not doing so in the context of this study weakens its quality. First, it is the norm to collect only the demographic information relevant to the research question. In this study, that was to examine whether SPS varied by exercise behavior and/or by intensity preference. The IRB at my institution requires a rationale for collecting demographic information that goes beyond what is necessary to provide a basic description of the sample (e.g., age, gender) and/or to address the stated purpose of your study. I did not set out to determine whether SPS varied by class level, major, and so on, and there is currently no literature to indicate SPS varies by these variables, so I had no rationale for including them. Those are interesting questions to explore in future research, but they were not relevant to my research questions. If research indicated class, major, and so on are related to SPS, or if I used a different sampling strategy, such as quota sampling, and then omitted key demographics of the sample, I could understand the criticisms. In this case, all a researcher needs to do replicate my study is use the same sampling strategy in a similar setting using the same questions. Listing the lack of additional demographic information as a limitation does not seem appropriate based on the purpose of my study and the lack of a rationale for including those variables. For reference, other investigators in the field of SPS research also provide limited demographic information when working with university populations, and what they do provide is often used only to describe the sample and is used limitedly in their analyses (e.g., https://doi.org/10.1038/s41398-017-0090-6 and https://doi.org/10.1016/j.paid.2018.01.047). This might be an area for improvement in research related to SPS, as it does raise questions about whether SPS influences other aspects of behavior, such as chosen major, exam performance, graduation rates, etc.
Comment 2: The authors conducted a Student's t-test and a one-way ANOVA. The effect size (e.g. Cohen's d) should be added to Table 1 and discussed in the description of the results above the table. Also, the effect size for the ANOVA (e.g. partial eta-square) is missing. Please, fill these gaps, by reporting the effect size and its interpretation appropriately. Response 2: Thank you for noticing this oversight. I have added Cohen's d to the table (p. 4-5, ln. 188-189) and it's interpretation to the paragraph ahead of it (p. 4, ln. 180-184). I have also added the eta squared for the ANOVA and its interpretation (p. 5. ln. 205-206).Reviewer 3 Report
Comments and Suggestions for Authors
Dear author,
thank you for addressing my comments.
Regards,
Author Response
Comments 1:
Dear author,
thank you for addressing my comments.
Regards,
Response 1: It was my pleasure. Thank you for taking the time to review my manuscript.